# CMIP5-Based Projection of Decadal and Seasonal Sea Surface Temperature Variations in East China Shelf Seas

**Huiqiang Lu [1], Chuan Xie [2], Cuicui Zhang [1,\*] and Jingsheng Zhai [1,\*]**

[1] School of Marine Science and Technology, Tianjin University, 92 WeiJin RD, Nankai District, Tianjin 300072, China; chentu90@163.com
[2] School of Oceanography, Shanghai Jiaotong University, 800 DongChuan RD, Minhang District, Shanghai 200240, China; xiechuan@sjtu.edu.cn
\* Correspondence: cuicui.zhang@tju.edu.cn (C.Z.); jingsheng@tju.edu.cn (J.Z.)

**Abstract:** The East China Shelf Seas, comprising the Bohai Sea, the Yellow Sea, and the shelf region of East China Sea, play significant roles among the shelf seas of the Western North Pacific Ocean. The projection of sea surface temperature (SST) changes in these regions is a hot research topic in marine science. However, this is a very difficult task due to the lack of available long-term projection data. Recently, with the high development of simulation technology based on numerical models, the model intercomparison projects, e.g., Phase 5 of the Climate Model Intercomparison Project (CMIP5), have become important ways of understanding climate changes. CMIP5 provides multiple models that can be used to estimate SST changes by 2100 under different representative concentration pathways (RCPs). This paper developed a CMIP5-based SST investigation framework for the projection of decadal and seasonal variation of SST in East China Shelf Seas by 2100. Since the simulation results of CMIP5 models may have degrees of errors, this paper uses hydrological observation data from World Ocean Atlas 2018 (WOA18) for model validation and correction. This paper selects seven representative ones including ACCESS1.3, CCSM4, FIO-ESM, CESM1-CAM5, CMCC-CMS, NorESM1-ME, and Max Planck Institute Earth System Model of medium resolution (MPI-ESM-MR). The decadal and seasonal SST changes in the next 100 years (2030, 2060, 2090) are investigated by comparing with the present analysis in 2010. The experimental results demonstrate that SST will increase significantly by 2100: the decadal SST will increase by about 1.55 °C, while the seasonal SST will increase by 1.03–1.95 °C.

**Keywords:** decadal and seasonal SST variation; East China Shelf Seas; CMIP5; WOA18

## 1. Introduction

The climate is changing. Our Earth is warming up. Many agree that climate change may be one of the greatest threats facing our planet. Ocean warming, which contributes much to global warming, has become a more serious problem. Recent observation-based estimates and model simulation results show that ocean warming is accelerating and will continue in the next one hundred years [1]. Since global warming may lead to many ecological problems, more efforts need to be made in the assessment and projection of the warming rate of the oceans in the future. The sea surface temperature (SST), which is an important physical parameter of oceans, can reflect the effect of climate change. The estimation of SST variation has become a hot research topic in marine science.

The East China Shelf Seas, which consist of the Bohai Sea, the Yellow Sea, and the shelf region of East China Sea, have been recognized as the most significant marginal seas in the Western North Pacific Ocean (WNPO) [2]. The Bohai Sea is a shallow semi-enclosed sea with an average depth of only 18 m, and the changes of its water temperature have a greater influence on its ecosystem than the physical forcing from the external oceans [3]. The Yellow Sea is a wide and shallow sea, where the depth in most regions is less than 50 m. Its water column can be seriously affected by the atmospheric conditions such as heating,

cooling, and wind stress within it than from the open seas [4]. The East China Sea, which is located between the largest continent and the largest ocean, has been largely effected by the climatic forcing from both the high-latitude Northern Hemisphere (East Asian Monsoon System) and the tropic ocean (Kuroshio Current (KC)) and generates characteristic climate patterns with strong horizontal and vertical temperature gradients [5–7]. These East China Shelf Seas are very sensitive to global warming because of their shallow waters [8]. The SST increase over these regions is about 0.8 to 2 °C per century, which is nearly twice the global average increase of SST [9]. The SST increases can not only affect the metabolic rates of marine organisms, but also influence other oceanic states, such as local currents [3]. Therefore, we need to analyze the SST variation, especially to project the long-term spatial and temporal variations in the future. However, this is a very difficult task because of the lack of effective long-term projection data [8].

Recently, with the rapid development of ocean simulations, model intercomparison projects have become important ways to investigate climate changes. Among them, the Coupled Climate Model Intercomparison Project plays an important role [10]. Phase 5 of the Climate Model Intercomparison Project (CMIP5) is an international collaboration framework, which provides a multimodel context to help understand the responses of climate models to a common forcing with the aim of promoting the climate model projection and assessment for the Fifth Assessment Report (AR5) of the Intergovernmental Panel on Climate Change (IPCC) [11]. CMIP5 contains multiple models, which can be used to project climate changes and sea level rise in the future under different climate change scenarios or representative concentration pathways (RCPs) including RCP2.7, RCP4.5, RCP6.0, and RCP8.5. These scenarios correspond to the peak of the atmosphere radiative imbalance of 2.6, 4.5, 6.0, and 8.5 $W/m^2$ by 2100, respectively [12]. Following the previous CMIPs, CMIP5 is a new experimental framework, which can be widely applied for the analysis of decadal and seasonal climate changes in the future.

In the literature, several CMIP5-based SST projection works have been developed. For example, Zhou and Ying [13] analyzed the interannual variability of the SST over the Pacific in the historical simulation and future analysis under RCP4.5 and RCP8.5. Qu and Huang [14] investigated the decadal variability of the tropical Indian Ocean SST–South Asian High (SAH) relation, as well as its response to global warming. Qin and Xie [15] studied the connections between the precipitation extremes during 1953–2002 in the dry and wet regions of China and the SST in the eastern tropical Pacific Ocean based on two sets of observation data, 17 CMIP5 models, and nine regional climate models. Tachibana et al. [16] examined the western Indian Ocean SST biases among the CMIP5 models and found that the multimodel ensemble mean SST biases over the western equatorial Indian Ocean are warmer than the observations during the summer monsoon season. Song et al. [17] evaluated 18 CMIP5 models according to their capability of simulating the SST annual cycle in the eastern equatorial Pacific. Xu et al. [18] tested a number of previously proposed mechanisms responsible for the southeastern tropical Atlantic SST bias based on CMIP5 models. Zhao and Zhang [19] analyzed the impacts of SST warming in the tropical Indian Ocean on the projected change in summer rainfall over Central Asia based on historical and RCP8.5 experiments. Kucharski and Joshi [20] evaluated the teleconnection from the tropical South Atlantic SST anomalies to the Indian monsoon based on observations and CMIP5 model data. Levine et al. [21] examined the extent and impact of cold SST biases developing in the northern Arabian Sea in the CMIP5 multi-model ensemble. Langehaug et al. [22] investigated the projection of SST in the Nordic Seas and Barents Sea using initialized hindcast simulations performed with three climate models (MPI-ESM-LR, CNRM-CM5, IPSL-CM5-LR). For the Chinese coastal seas, a few works have also been presented to study the SST changes using CMIP5 models. For example, Huang et al. [23] evaluated the capacities of 17 selected CMIP5 models on the historical SST simulation in the South China Sea and projected the SST changes in the 21st Century under RCP2.6, RCP 4.5, and RCP8.5, respectively. Tan et al. [24] evaluated the variation trend of SST over offshore China in the 21st Century based on the selected CMIP5

models under RCP4.5. Song et al. [25] assessed the monthly, seasonal, and interannual SSTs in the China seas over 1960–2002 using five representative CMIP5 models.

Although existing works have made some achievements in the study of the SST changes in the East China Shelf Seas, there are still some problems unsolved. Firstly, existing works mainly focused on the study of annual and decadal variations of the SST in the East China Shelf Seas without investigating the seasonal variations, which may affect ocean organisms and the ecological environment more seriously (e.g., the changes of species distribution and the move up of the phenophase). Secondly, existing works lack precise ocean observation data for model validation. Existing works mainly used the low-resolution observation data HadISSTfrom the Hadley Center in the U.K. ($1° \times 1°$). It is not accurate enough for the evaluation of high-resolution CMIP5 models (e.g., the Max Planck Institute Earth System Model of medium resolution (MPI-ESM-MR) of $0.1° \times 0.1°$). Moreover, existing works only used the observation data for model validation without establishing model correction or simulation result modification, which is a fundamental procedure to make projections more accurate. Thirdly, the models used in existing works are not very appropriate. Several high-resolution models (e.g., the MPI-ESM-MR) were not fully evolved in the existing works. Fourthly, the study region of existing works was very rough. Existing works mainly referred to the Chinese coastal seas as the rectangle region of 0° N–45° N, 100° E–140° E, which is a very coarse region to study on the SST changes in the East China Shelf Seas.

To solve these problems, this paper establishes an SST analysis framework, which can be used for the projection of both decadal and seasonal SST variation in the East China Shelf Seas using high-resolution CMIP5 data. This paper investigates both the decadal and seasonal SST variation. For the second problem, we introduce the high-resolution hydrological observation data WOA18 ($0.25° \times 0.25°$) from World Ocean Atlas 2018 [26] for model validation and calculate the error map of each of the CMIP5 model data for the simulation result modification. This strategy ensures the high accuracy of CMIP5 model data. To solve the third problem, we select the best model MPI-ESM-MR with the highest resolution ($0.1° \times 0.1°$) from seven representative models (ACCESS1.3, CCSM4, FIO-ESM, CESM1-CAM5, CMCC-CMS, NorESM1-ME, and MPI-ESM-MR). MPI-ESM-MR not only has low errors, but its high resolution also guarantees the specific analysis of local climate changes. To solve the fourth problem, we utilize the bathymetric depth map with a finer grid to define the study area (as shown in Figure 1). In this paper, the decadal and seasonal SST variation analysis is performed by comparing the SST simulation result on 2030, 2060, and 2090 with the present analysis in 2010 under RCP4.5. We use RCP4.5 as a representative scenario because it is a medium-mitigation emission scenario that stabilizes direct radiative forcing at 4.5 W/m$^2$ (650 ppm CO$_2$ equivalent) by 2100 [27,28]. Compared to RCP4.5, RCP2.6 is a mitigation scenario leading to a very low forcing level (at 2.6 W/m$^2$ by 2100), while RCP8.5 is a scenario with very high greenhouse gas emissions (at 8.5 W/m$^2$ by 2100). RCP4.5 is neither too high, nor too low for projection. To the best of our knowledge, this is the first work to investigate both decadal and seasonal SST variations in the East China Shelf Seas by 2100 using high-resolution CMIP5 data.

This paper is organized as follows: Section 2 introduces the data and the study area. The model validation and simulation result modification are performed by comparing the CMIP5 model data with WOA18 data in Section 3. Sections 4 and 5 analyze the projection results of decadal and seasonal SST variations, respectively. Section 6 concludes this paper.

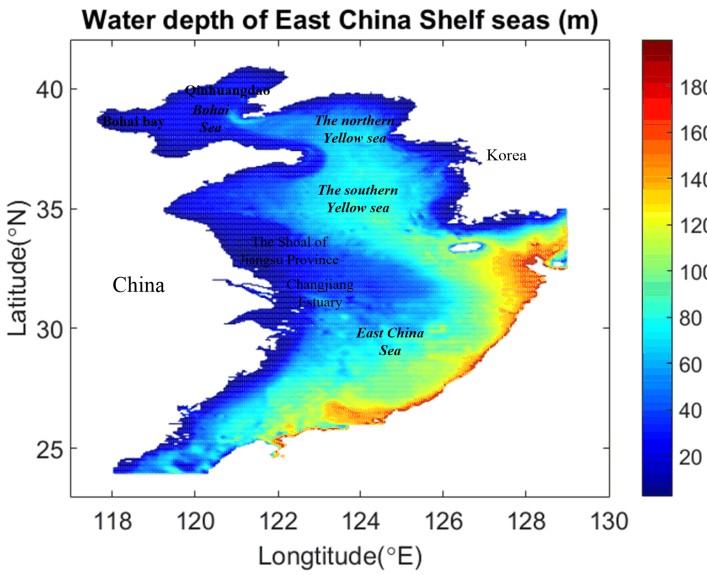

**Figure 1.** Depth map of East China Shelf Seas.

## 2. Data and Study Region

In this paper, CMIP5 model data were used for the projection of future SST changes, while WOA18 data were used for model validation, as well as for present analysis. CMIP5 consists of 36 models, which are developed by 16 institutes. In the preprocessing procedure, we selected the most representative models according to the following criteria: (1) for each institute, we selected the latest developed, with the highest resolution model for study; (2) the selected models should provide the simulation results of the SST from 2010 to 2100 continuously. We finally chose seven representative models. Their information is shown in Table 1. From Table 1, we can see that the resolution of Max Planck Institute Earth System Model of medium resolution (MPI-ESM-MR) is much higher than that of other models. MPI-ESM-MR couples general circulation models for the ocean and the atmosphere. It has been widely applied in many climate change experiments for either idealized $CO_2$-only forcing or forcings based on observations and RCP scenarios [29].

**Table 1.** CMIP5 models used in this paper: model name, average horizontal resolution (latitude × longitude), and reference. MPI-ESM-MR, Max Planck Institute Earth System Model of medium resolution.

| Model Name | Horizontal Resolution | Reference |
|---|---|---|
| ACCESS1.3 | 0.3° × 1.0° | Bi et al., 2013 [30] |
| CCSM4 | 0.5° × 1.1° | Danabasoglu et al. 2012 [31] |
| FIO-ESM | 0.5° × 1.1° | Qiao et al. 2013 [32] |
| CESM1-CAM5 | 0.9° × 1.25° | Meehl et al. 2013 [33] |
| CMCC-CMS | 0.5° × 2.0° | Borrelli et al. 2012 [34] |
| NorESM1-ME | 0.5° × 1.1° | Schwinger et al. 2016 [35] |
| MPI-ESM-MR | 0.1° × 0.1° | Jungclaus et al. 2013 [36] |

The annual and seasonal SST data in 2010 were obtained from the World Ocean Atlas 2018 (WOA18) hydrological observation averaged over 2005 to 2017. WOA18 provides both objectively analyzed (1° grid) climatological fields of in situ temperature for annual and seasonal compositing periods of the World Ocean. It also includes associated statistical observation data interpolated on 5°, 1°, and 0.25° grids [26]. We used the highest resolution data of 0.25° grid. As shown in Figure 1, the study area is comprised of the Bohai Sea, the Yellow Sea, and the shelf region of East China Sea.

Since each selected CMIP5 model has its inner simulation variations and errors, we calculated the climatological annual and seasonal water temperature in 2010 by the average result over 2006 to 2015. A similar processing was done for 2030, 2060, and 2090. They were investigated by the average result of 2026~2035, 2056~2064, and 2086~2094, respectively.

### 3. Model Validation and Simulation Result Modification

As we know, the simulation results of numeric ocean models may contain some degree of errors. Before using the CMIP5 model data, we needed to perform model validation to evaluate whether the simulation results are accurate enough for SST projection. The model validation was performed by comparing the annual and seasonal CMIP5 model data with the hydrological observation data of WOA18 in 2010. The qualitative and quantitative comparison results are shown in Figure 2 and Table 2, respectively. From Figure 2, we can find that most selected CMIP5 models have a similar SST distribution as the WOA18 data. However, different models also vary from each other in the model resolution and simulation accuracy. Among these models, the simulation result of MPI-ESM-MR is better than the others. Its resolution (in 0.1°) is the highest one (even higher than WOA18), and its simulation result is most similar to WOA18. MPI-ESM-MR can illustrate the local climate impact factors such as the Yellow Sea cold water mass in Summer and Autumn, the Yellow Sea warm current in Winter, the Kuroshio invasion, and the northern flow of the Taiwan warm current around Changjiang shore to the Tsushima Strait. Benefiting from the high resolution, its simulation result is even better than the observation data (WOA18). The numerical results in Table 2 also verify this finding. From Table 2, we can see that both the annual and seasonal simulation errors of MPI-ESM-MR are low enough. That means that MPI-ESM-MR is the best model that can be used for the projection of decadal and seasonal SST variations.

**Table 2.** Comparison of CMIP5 model data with the real observation data from WOA18 in 2010.

| Data Source | | Annual | Spring | Summer | Autumn | Winter |
|:---:|:---:|:---:|:---:|:---:|:---:|:---:|
| WOA18 | | 18.07 | 16.80 | 25.79 | 18.49 | 11.20 |
| CMIP5 Model Errors | ACCESS1.3 | 0.67 | 1.49 | 2.42 | 0.61 | 0.62 |
| | CCSM4 | 0.33 | 0.77 | 0.14 | 1.51 | 0.72 |
| | FIO-ESM | 0.79 | 0.17 | -0.12 | 1.67 | 1.44 |
| | CESM1-CAM5 | 0.28 | 0.85 | 0.64 | 1.55 | 1.05 |
| | CMCC-CMS | 0.40 | −0.50 | −0.66 | 1.61 | 1.17 |
| | NorESM1-ME | 0.52 | −0.40 | −0.05 | 1.80 | 0.75 |
| | MPI-ESM-MR | 0.15 | −0.96 | 0.81 | 1.67 | 0.72 |
| | Average | 0.26 | −0.71 | −0.70 | 1.52 | 0.94 |

After model validation, we performed simulation result modification to make the CMIP5 model data more accurate. Firstly, we calculated the error map of MPI-ESM-MR by comparing its results with WOA18 data for the year and the four seasons of 2010, respectively. Then, we subtracted these errors from the original simulation results to obtain more accurate projection data. Figure 3 illustrates the annual and seasonal error maps of MPI-ESM-MR. From Figure 3a–e, we can see that the error of MPI-ESM-MR is mainly concentrated in the deep trough of the Yellow Sea, where its simulation on the influence of the Yellow Sea warm current is stronger than the observation result. This phenomenon leads to the higher SST simulation in the eastern Yellow Sea in spring and winter and in the northern part of East China Shelf Seas in autumn than the WOA18 data. However, its simulation results of other regions were satisfactory enough for projection.

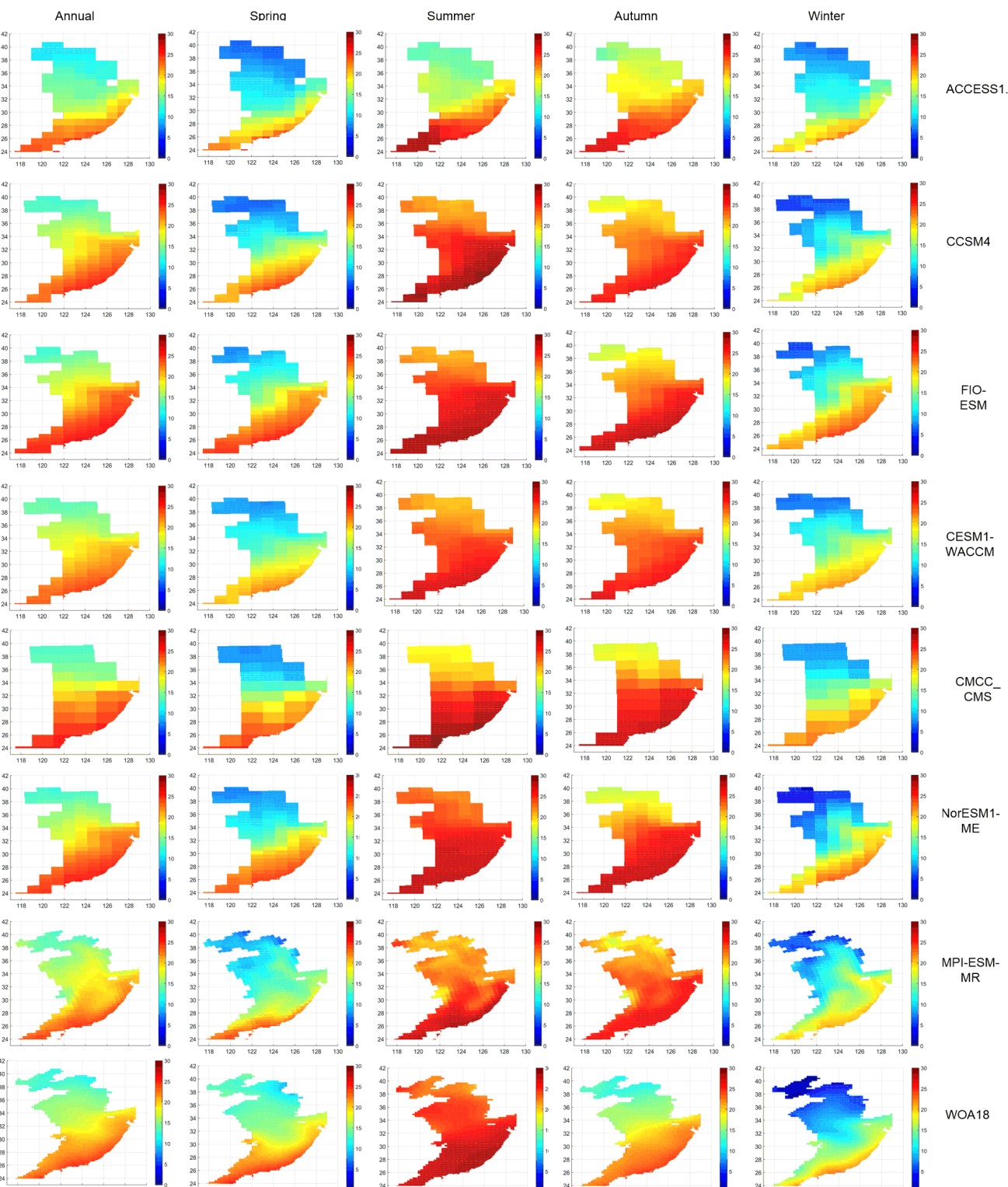

**Figure 2.** Model validation: the comparison of CMIP5 simulation results with WOA18. The first row to seventh row illustrate the result of seven CMIP5 models: ACCESS1.3, CCSM4, FIO-ESM, CESM1-CAM5, CMCC-CMS, NorESM1-ME, and MPI-ESM-MR, respectively. The last row shows the observation result from WOA18 data. The first to fifth column present the results in the year round, spring, summer, autumn, and winter of 2010, respectively.

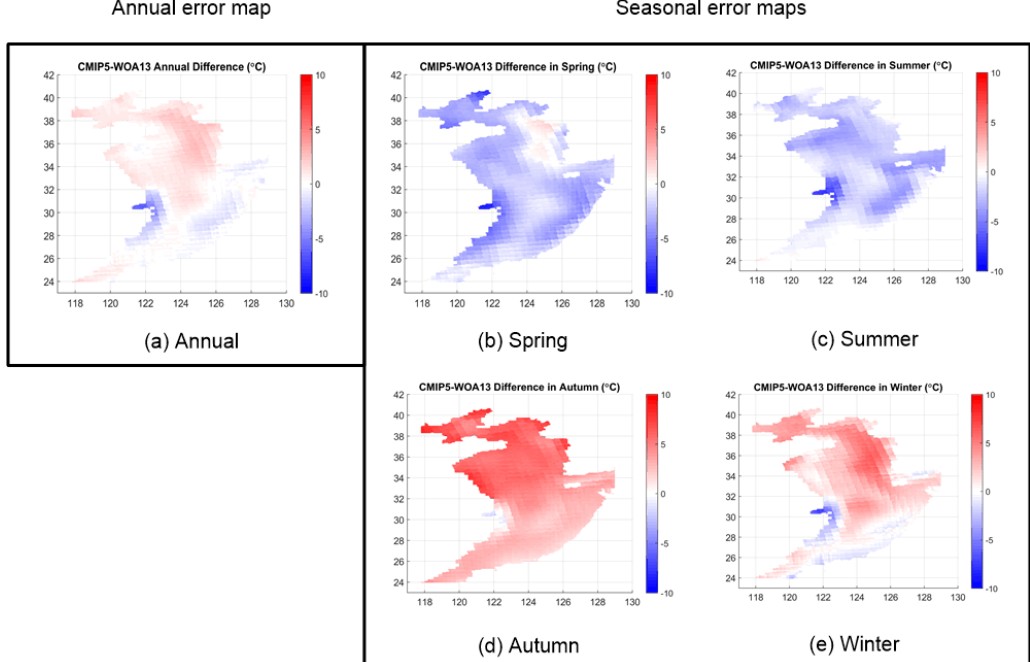

**Figure 3.** The error maps of MPI-ESM-MR in 2010: (**a**) annual, (**b**) spring, (**c**) summer, (**d**) autumn, and (**e**) winter.

## 4. Projection Results of Decadal SST Changes

We analyzed the decadal SST changes in 2030, 2060, and 2090 compared with the present analysis in 2010. The modified projection results of the seven models and their average results are shown in Table 3 and Figure 4, respectively. From Table 3 and Figure 4, we find that most models demonstrate a significant SST increase in the next 100 years. From Table 3, we also find that the simulation results of MPI-ESM-MR are most similar to the average of all models. Moreover, its resolution is much higher than that of other models. Since the resolution of the other six models is too low to distinguish local climate impact factors such as the Kuroshio system, it is inappropriate to use them to project the SST changes in the future. Therefore, next, we used the modified result of MPI-ESM-MR to analyze the decadal SST changes. From Table 3, we can see that the SST increases from 18.09 °C in 2010 to 18.68 °C in 2030, 19.78 °C in 2060, and 19.65 °C in 2090. In particular, a remarkable SST increase was obtained from 2030 to 2060, which was 1.10 °C. Till 2090, the SST will increase by 1.55 °C. Besides the decadal analysis, we also investigated the annual changing rate of SST from 2010 to 2090. The changing rate between 2030 and 2010, 2060 and 2030, and 2090 and 2060 is illustrated in Table 3. We can see that this rate is very small from 2010 to 2030, achieves the top value from 2030 to 2060, and becomes low from 2060 to 2090. This suggests that the SST increase becomes stable from 2060 to 2090.

Then, we analyzed the spatial SST changes in different regions through the qualitative results shown in Figure 5. Figure 5a,c,e shows the SST projection in 2030, 2060, and 2090 and their changes compared to 2010 in (b), (d), and (f), respectively. From Figure 5, we can find that the highest SST increase rate is obtained in 2060 (in Figure 5d), and the increase becomes stable in 2090 (in Figure 5f). Although the SST increase in 2030 (in Figure 5b) is not as significant as that in 2060, its distribution is inhomogeneous. The significant SST increases mainly concentrate in the east of Bohai (especially east of Qinhuangdao), the Changjiang Estuary, and its adjacent shore of Jiangsu Province. The increment can reach to as high as 1.5 °C. From Figure 5d, we can see that compared to 2030, the increase of the SST in 2060 expands to almost all the regions and at an even higher rate. The increment reaches about 3.5 °C in the Changjiang Estuary and in the outer shelf of the East China Sea. From Figure 5f, we can see that the SST variation in 2090 is similar to that of 2060 and

becomes stable gradually. One remarkable thing to note is that the SST in the northern and central Yellow Sea is relatively lower than that in the other regions (see Figure 5a,c,e), and their SST increases are not very obvious (see Figure 5b). This is mainly due to the presence of the cold water mass in the Northern and Central Yellow Seas, which has a great influence on the distribution of the SST. The increase of the SST can severely influence marine ecosystems and cause many ecological problems, especially for shelf regions with shallow water. For example, the Brown tide has broken out in the Qinhuangdao coastal area of Bohai recurrently since 2009. The Changjiang Estuary and the shore in Northern Jiangsu Province suffered from the Red Tide and Green Tide, and jellyfish disasters often have also occurred in these areas recently. The increase of the SST will intensify these problems through the following mechanisms. First, it can affect the metabolic rate of marine organisms. Second, it can influence the other oceanic states, such as local currents. Third, it can further affect the water column stratification, substrate structure, photosynthetic light intensity, and nutrient cycling [3]. The species distribution can also be disturbed by the increase of the SST. For example, the warm and cold water fish stocks can be reduced greatly in the regions with significant SST increases, and tropical ocean organisms may move up to the middle and high latitude areas with warm water [37].

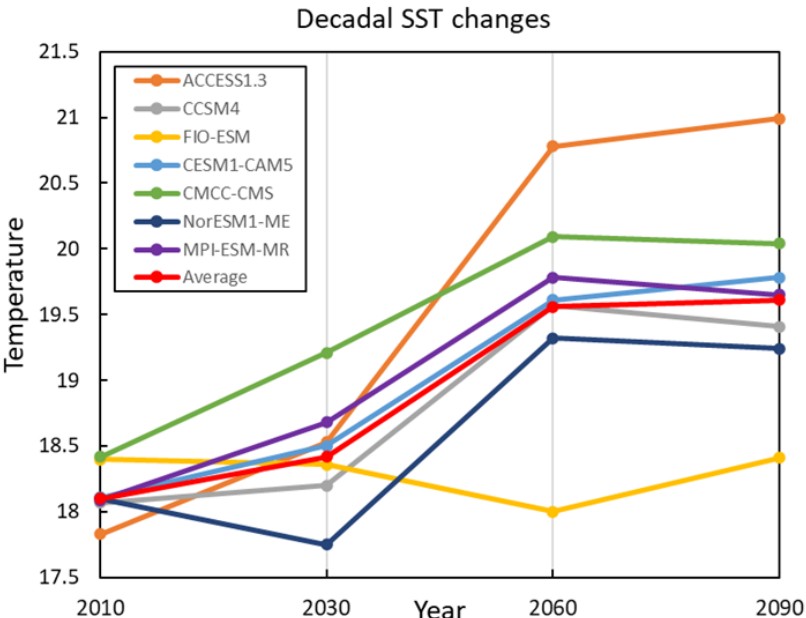

**Figure 4.** The decadal SST variations in the next 100 years.

**Table 3.** The decadal SST variations in the next 100 years.

| SST | 2010 | 2030 | 2060 | 2090 |
|---|---|---|---|---|
| ACCESS1.3 | 17.83 | 18.53 | 20.78 | 20.99 |
| CCSM4 | 18.07 | 18.20 | 19.57 | 19.41 |
| FIO-ESM | 18.40 | 18.36 | 18.00 | 18.41 |
| CESM1-CAM5 | 18.11 | 18.50 | 19.61 | 19.78 |
| CMCC-CMS | 18.42 | 19.21 | 20.09 | 20.04 |
| NorESM1-ME | 18.10 | 17.75 | 19.32 | 19.24 |
| MPI-ESM-MR | 18.09 | 18.68 | 19.78 | 19.65 |
| Average | 18.10 | 18.42 | 19.56 | 19.61 |



**Table 3.** *Cont.*

| SST Changes | 2010 | 2030–2010 | 2060–2010 | 2090–2010 |
| --- | --- | --- | --- | --- |
| | | (2030–2010)/Years | (2060–2030)/Years | (2090–2060)/Years |
| ACCESS1.3 | 17.83 | 0.70 | 2.95 | 3.16 |
| | | 0.033 | 0.073 | 0.007 |
| CCSM4 | 18.07 | 0.13 | 1.50 | 1.34 |
| | | 0.006 | 0.044 | −0.005 |
| FIO-ESM | 18.40 | −0.04 | −0.40 | 0.02 |
| | | −0.002 | −0.011 | 0.013 |
| CESM1-CAM5 | 18.11 | 0.40 | 1.50 | 1.67 |
| | | 0.019 | 0.036 | 0.005 |
| CESM1-CMS | 18.42 | 0.79 | 1.67 | 1.62 |
| | | 0.038 | 0.028 | −0.002 |
| NorESM1-ME | 18.10 | −0.35 | 1.22 | 1.14 |
| | | −0.016 | 0.051 | −0.003 |
| MPI-ESM-MR | 18.10 | 0.59 | 1.68 | 1.55 |
| | | 0.028 | 0.035 | −0.004 |
| Average | 18.11 | 0.32 | 1.46 | 1.51 |
| | | 0.014 | 0.036 | 0.002 |

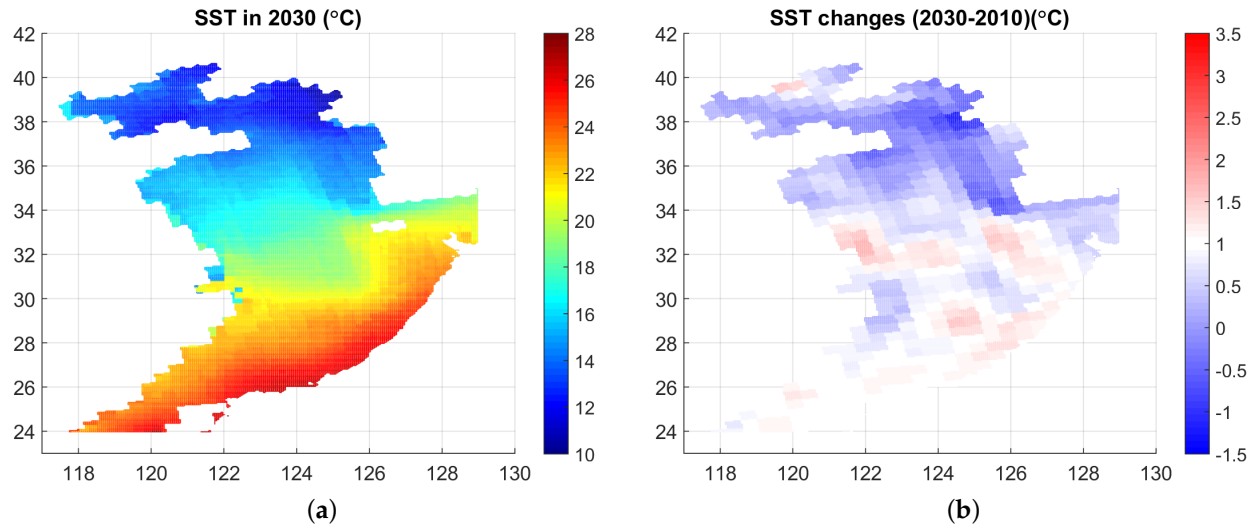

(a)

(b)

**Figure 5.** *Cont.*

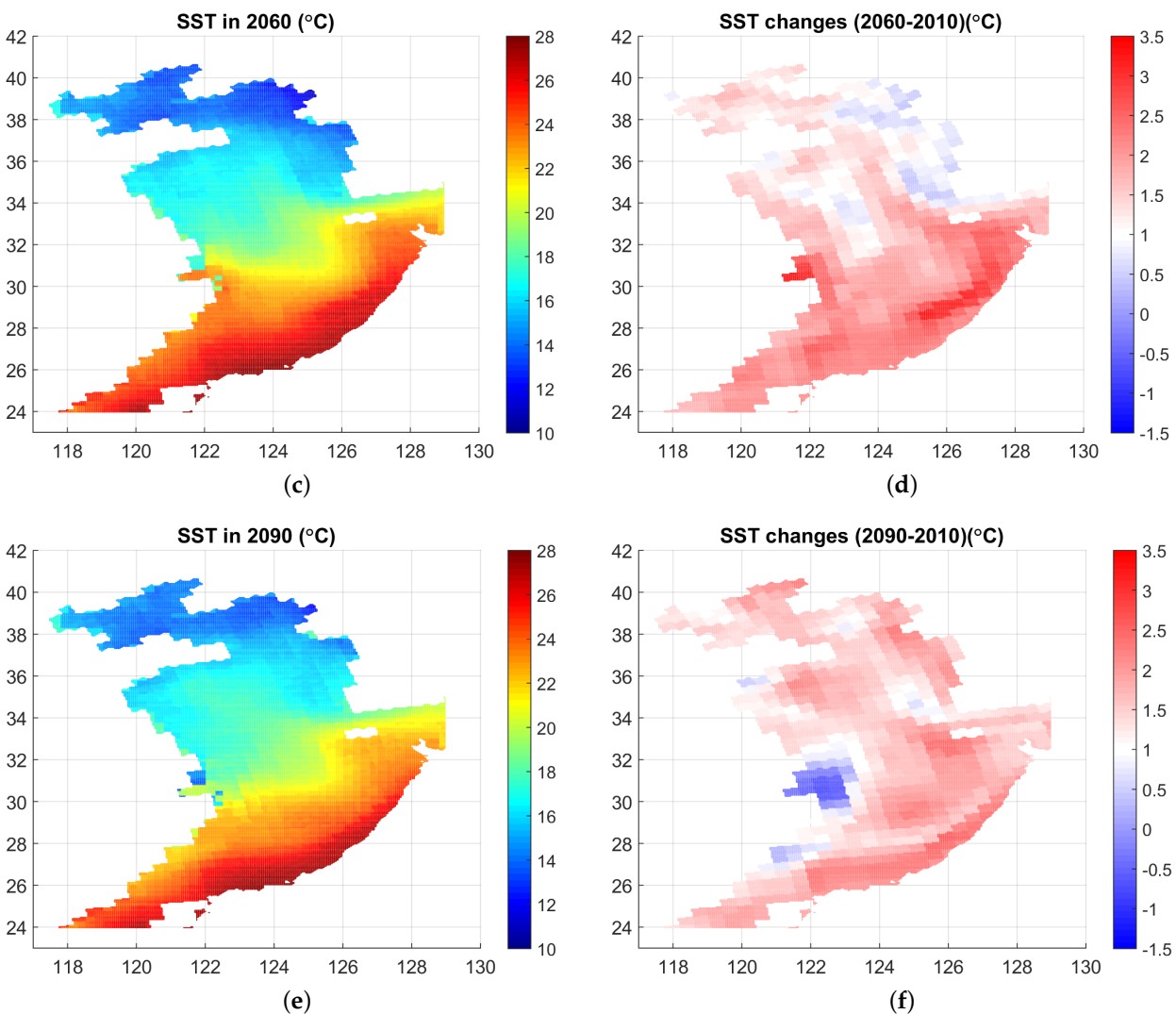

**Figure 5.** The projection results of decadal SST changes by MPI-ESM-MR in (**a**) 2030, (**c**) 2060, and (**e**) 2090 and their changes compared to that in 2010 in (**b**) 2030–2010, (**d**) 2060–2010, and (**f**) 2090–2010.

## 5. Projection Results of Seasonal SST Changes

We analyzed the seasonal SST variations in the next 100 years. The numerical results of the seven models and their average results are shown in Table 4. Similar to the decadal analysis, we used the modified simulation result of MPI-ESM-MR to investigate the seasonal SST changes because of its high resolution and low projection error. From Table 4, we can see that the SST increases significantly from 2010 to 2090 for all seasons. The increment reaches about 2 °C by 2090. Next, we analyzed the spatial seasonal SST variations. The qualitative simulation results are shown in Figure 6. The first to fourth rows show the SST changes from 2010 to 2030, 2060, and 2090 in spring, summer, autumn, and winter, respectively. We found that in summer, the regions with significant SST increases mainly are concentrated in the east of Qinhuangdao in Bohai, the shore of Northern Jiangsu Province in the Yellow Sea, and the Changjiang Estuary in the East China Sea, which is coincident with that in the decadal analysis. This is mainly because the upwelling waters become weak and the Kuroshio invasion becomes stronger in summer. In autumn, the increase of the SST becomes smoother, and the regions with a significant SST increase move from the inner shelf area to the middle and even outer shelf regions in winter. In particular, the outer shelf regions have higher SST increases than other regions. A remarkable demonstration of the SST increase in the Northern Yellow Sea is illustrated in Figure 6f, where the Yellow

Sea warm current plays a more significant role in summer than the Yellow Sea cold water mass. The SST increases in this region will intensify the low oxygen and acidification problems. Similar to the decadal analysis, for all seasons, the SST increase in 2060 (shown in the second column) is more significant than that in 2030 (shown in the first column) and becomes stable in 2090 (shown in the third column). As previously mentioned, the increase of the SST may cause many environmental problems and destroy the marine ecosystem. For example, the increase of the SST in Bohai may lead to the Brown Tide problem, and the SST increases in the shore of Jiangsu Province and the Changjiang Estuary may result in the Red and Green Tide problem and even cause the outbreak of jellyfish disasters in these regions. The increase of the SST in the middle shelf and outer shelf regions of the East China Sea will intensify the marine acidification and ocean hypoxia there. The model validation and modification of this paper is relative simple. We use the difference between the CMIP5 model and the WOA data in 2010 as the model error and substrate these errors from CMIP5 model data to modify their projection results. We should use more sophisticated methods with all the model assumptions and conditions to perform model modification. However, it is not an easy task especially for the future projection with many unknown conditions. We donot know whether these errors will remain constant for the 2010–2100 period. Thus. we include the experimental results without projection modification for comparison in this part. The decadal and seasonal SST changes without modification of model projection results are shown in Table A1 and Table A2, respectively. Comparing Table A1 to Table 3, and Table A2 to Table 4, we find that the bottom part of these tables are the same with each other. That is, the investigation of 35 the SST changes in the future using the differences of 2030, 2060, and 2090 relative to that in 2010 is not influenced by the 36 model validation and modification method in this paper.

**Table 4.** The projection results of seasonal SST variations in the next 100 years using seven CMIP5 models and their average result.

| SST | 2010 | | | | 2030 | | | | 2060 | | | | 2090 | | | |
|---|---|---|---|---|---|---|---|---|---|---|---|---|---|---|---|---|
| | Spr. | Sum. | Aut. | Win. | Spr. | Sum. | Aut. | Win. | Spr. | Sum. | Aut. | Win. | Spr. | Sum. | Aut. | Win. |
| ACCESS1.3 | 16.20 | 24.86 | 18.77 | 11.48 | 16.55 | 25.38 | 19.63 | 12.57 | 19.47 | 28.28 | 21.75 | 13.64 | 18.78 | 28.49 | 22.65 | 14.05 |
| CCSM4 | 16.43 | 25.71 | 18.85 | 11.29 | 16.45 | 25.36 | 19.57 | 11.44 | 18.04 | 26.61 | 20.53 | 13.08 | 17.83 | 26.64 | 20.68 | 12.49 |
| FIO-ESM | 16.94 | 25.84 | 19.06 | 11.75 | 16.36 | 26.20 | 19.26 | 11.60 | 15.52 | 25.95 | 19.78 | 10.77 | 16.45 | 26.68 | 19.07 | 11.47 |
| CESM1-CAM5 | 16.48 | 25.64 | 18.88 | 11.42 | 17.33 | 26.21 | 19.03 | 11.45 | 18.65 | 27.36 | 19.98 | 12.46 | 17.94 | 27.54 | 21.02 | 12.61 |
| CMCC-CMS | 16.76 | 25.60 | 19.32 | 11.99 | 18.10 | 26.42 | 19.58 | 12.75 | 18.83 | 27.64 | 20.46 | 13.44 | 18.83 | 27.39 | 20.10 | 13.85 |
| NorESM1-ME | 16.54 | 25.77 | 18.85 | 11.24 | 16.80 | 25.73 | 18.18 | 10.30 | 17.93 | 27.68 | 19.72 | 11.96 | 17.96 | 27.36 | 19.52 | 12.13 |
| MPI-ESM-MR | 16.66 | 25.71 | 18.73 | 11.26 | 17.36 | 26.88 | 18.41 | 12.06 | 19.17 | 27.12 | 19.71 | 13.10 | 18.27 | 27.76 | 19.76 | 12.88 |
| Average | 16.67 | 25.71 | 18.72 | 11.31 | 17.08 | 26.15 | 18.88 | 11.58 | 18.34 | 27.35 | 20.07 | 12.48 | 18.11 | 27.53 | 20.19 | 12.61 |

| SST Changes | 2010 | | | | 2030–2010 | | | | 2060–2010 | | | | 2090–2010 | | | |
|---|---|---|---|---|---|---|---|---|---|---|---|---|---|---|---|---|
| | Spr. | Sum. | Aut. | Win. | Spr. | Sum. | Aut. | Win. | Spr. | Sum. | Aut. | Win. | Spr. | Sum. | Aut. | Win. |
| ACCESS1.3 | 16.20 | 24.86 | 18.77 | 11.48 | 0.35 | 0.52 | 0.86 | 1.09 | 3.27 | 3.41 | 2.97 | 2.16 | 2.58 | 3.63 | 3.87 | 2.57 |
| CCSM4 | 16.43 | 25.71 | 18.85 | 11.29 | 0.02 | −0.36 | 0.72 | 0.15 | 1.61 | 0.90 | 1.68 | 1.79 | 1.40 | 0.93 | 1.82 | 1.20 |
| FIO-ESM | 16.94 | 25.84 | 19.06 | 11.75 | −0.57 | 0.36 | 0.20 | −0.15 | −1.42 | 0.11 | 0.72 | −0.99 | −0.49 | 0.84 | 0.00 | −0.28 |
| CESM1-CAM5 | 16.48 | 25.64 | 18.88 | 11.42 | 0.85 | 0.57 | 0.15 | 0.03 | 2.16 | 1.72 | 1.10 | 1.04 | 1.45 | 1.90 | 2.14 | 1.19 |
| CMCC-CMS | 16.76 | 25.60 | 19.32 | 11.99 | 1.33 | 0.81 | 0.26 | 0.75 | 2.07 | 2.04 | 1.15 | 1.45 | 2.06 | 1.79 | 0.79 | 1.86 |
| NorESM1-ME | 16.54 | 25.77 | 18.85 | 11.24 | 0.27 | −0.05 | −0.68 | −0.93 | 1.39 | 1.91 | 0.87 | 0.73 | 1.42 | 1.59 | 0.67 | 0.89 |
| MPI-ESM-MR | 16.66 | 25.71 | 18.73 | 11.26 | 0.70 | 1.17 | −0.33 | 0.80 | 2.51 | 1.41 | 0.98 | 1.84 | 1.61 | 1.95 | 1.03 | 1.62 |
| Average | 16.67 | 25.71 | 18.72 | 11.31 | 0.41 | 0.44 | 0.16 | 0.26 | 1.67 | 1.64 | 1.35 | 1.16 | 1.45 | 1.82 | 1.47 | 1.29 |

| SST Changes ratios | 2010 | | | | 2030–2010 | | | | 2060–2030 | | | | 2090–2060 | | | |
|---|---|---|---|---|---|---|---|---|---|---|---|---|---|---|---|---|
| | Spr. | Sum. | Aut. | Win. | Spr. | Sum. | Aut. | Win. | Spr. | Sum. | Aut. | Win. | Spr. | Sum. | Aut. | Win. |
| ACCESS1.3 | 16.20 | 24.86 | 18.77 | 11.48 | 0.017 | 0.025 | 0.041 | 0.052 | 0.094 | 0.093 | 0.068 | 0.035 | 0.022 | 0.007 | 0.029 | 0.013 |
| CCSM4 | 16.43 | 25.71 | 18.85 | 11.29 | 0.001 | −0.017 | 0.034 | 0.007 | 0.051 | 0.041 | 0.031 | 0.031 | 0.053 | 0.007 | 0.008 | 0.005 |
| FIO-ESM | 16.94 | 25.84 | 19.06 | 11.75 | −0.027 | 0.017 | 0.009 | −0.007 | −0.027 | −0.008 | 0.017 | −0.027 | −0.030 | 0.024 | −0.023 | 0.023 |
| CESM1-CAM5 | 16.48 | 25.64 | 18.88 | 11.42 | 0.040 | 0.027 | 0.007 | 0.001 | 0.042 | 0.037 | 0.031 | 0.033 | 0.023 | 0.006 | 0.034 | 0.005 |
| CMCC-CMS | 16.76 | 25.60 | 19.32 | 11.99 | 1.33 | 0.81 | 0.26 | 0.75 | 2.07 | 2.04 | 1.15 | 1.45 | 2.06 | 1.79 | 0.79 | 1.86 |
| NorESM1-ME | 16.54 | 25.77 | 18.85 | 11.24 | 0.013 | −0.002 | 0.032 | 0.045 | 0.036 | 0.063 | 0.050 | 0.054 | 0.008 | −0.010 | −0.007 | 0.005 |
| MPI-ESM-MR | 16.66 | 25.71 | 18.73 | 11.26 | 0.034 | 0.056 | −0.016 | 0.038 | 0.058 | 0.008 | 0.042 | 0.034 | −0.029 | 0.017 | 0.002 | −0.007 |
| Average | 16.67 | 25.71 | 18.72 | 11.31 | 0.020 | 0.021 | 0.007 | 0.013 | 0.041 | 0.039 | 0.038 | 0.029 | −0.007 | 0.006 | 0.004 | 0.004 |

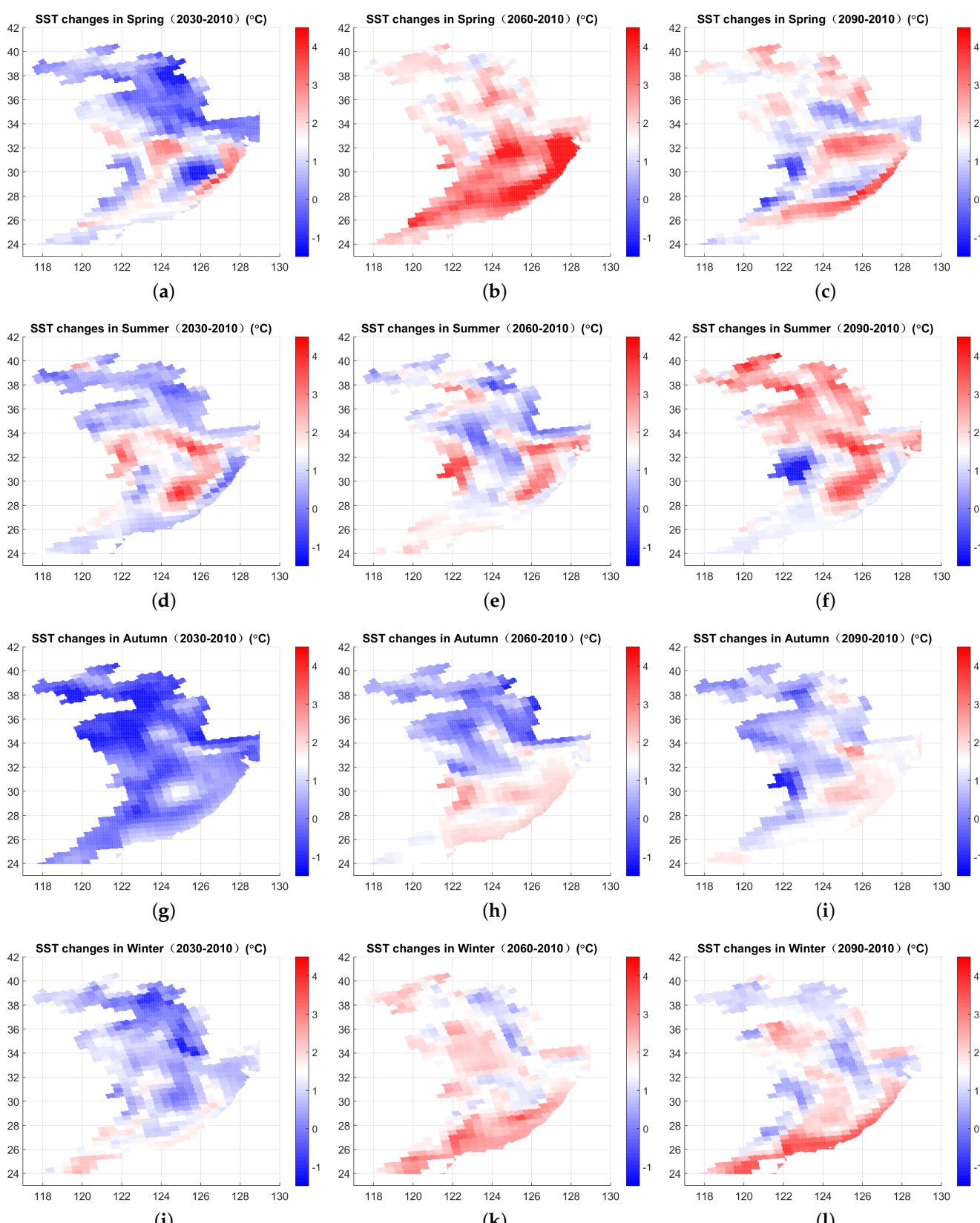

**Figure 6.** Seasonal changes of SST from 2010 to 2030 (the first column (**a**,**d**,**g**,**j**)), 2060 (the second column (**b**,**e**,**h**,**k**)), and 2090 (the third column (**c**,**f**,**i**,**l**)), respectively. The first row to fourth row illustrates the changes in Spring, Summer, Autumn, and Winter, respectively.

## 6. Conclusions

This paper investigated both decadal and seasonal SST variations in the East China Shelf Seas by 2100 using CMIP5 models under RCP4.5. Seven representative CMIP5 models were used in this paper, including ACCESS1.3, CCSM4, FIO-ESM, CESM1-CAM5, CMCC-CMS, NorESM1-ME, and MPI-ESM-MR. To make the projection more accurate, we introduced the high-resolution hydrological observation data from WOA18 for model validation and simulation result modification. The best model (MPI-ESM-MR) with highest resolution and low errors was selected for the projection of SST changes. The extensive experimental results demonstrated that both decadal and seasonal SSTs in the East China Shelf Seas will increase significantly in the next one hundred years: the decadal SST will increase by about 1.5 °C till 2090, and the seasonal SST will increase by about 1.03 °C–1.95 °C by 2090. The highest increment was obtained from 2030 to 2060, and it became stable from 2060 to 2090. Although the SST increase in 2030 was not as significant as that in 2060 and 2090, the distribution of SST increases was inhomogeneous. Some local regions had a high SST increase of 1.5 °C in the east of Bohai Sea (east of Qinhuangdao), the Changjiang Estuary, and its adjacent shore of Jiangsu Province. Compared to the decadal analysis, the seasonal variation may play a more important role in understanding climate changes. We found that in summer, the significant SST increase mainly concentrated in the eastern area of Bohai, the Changjiang Estuary, and the shore of Jiangsu Province, which is mainly due to the weakness of upwelling, the strengthening of the Kuroshio invasion, and the influence of the Taiwan warm current. In autumn, the increase of SST becomes smoother, and the regions with a significant SST increase move from the inner shelf to the outer shelf regions in winter and spring. The significant SST increase may affect the local marine ecology seriously. It can change the distribution of marine organisms, lead to the moving up of the phenophase, intensify the low oxygen and acidification problems, and cause many environmental problems. It has brought about the Brown Tide in the east of Qinhuangdao in the Bohai Sea, led to the Red and Green Tide in the Changjiang Estuary and the shore of Jiangshu Province, and caused the outbreak of jellyfish disasters in these regions. To the best of our knowledge, this is the first work to investigate both decadal and seasonal SST variations in the East China Shelf Seas using high-resolution CMIP5 model data and observation data.

**Author Contributions:** Conceptualization, C.Z.; Methodology, C.Z., C.X., H.L. and J.Z.; Analysis, C.Z., C.X. and H.L.; Writing and Editing, C.Z., H.L.; Project Administration, C.Z. and J.Z. All authors have read and agreed to the published version of the manuscript.

**Funding:** This research was funded by the National Natural Science Foundation of China Grant 41806116 and the National Key Research and Development Program of China 2016YFC1401203.

**Institutional Review Board Statement:** Not applicable.

**Informed Consent Statement:** Not applicable.

**Data Availability Statement:** Data used can be accesses via CMIP5 website.

**Acknowledgments:** H.L. acknowledges the National Natural Science Foundation of China Grant 41806116 and the National Key Research and Development Program of China 2016YFC1401203.

**Conflicts of Interest:** The authors declare no conflict of interest.

## Appendix A

**Table A1.** The decadal SST variation in the next 100 years using the raw CMIP5 model data without projection result modification.

| SST | 2010 | 2030 | 2060 | 2090 |
|---|---|---|---|---|
| ACCESS1.3 | 16.18 | 16.89 | 19.14 | 19.35 |
| CCSM4 | 18.93 | 19.06 | 20.42 | 20.26 |
| FIO-ESM | 20.48 | 20.44 | 20.08 | 20.50 |
| CESM1-CAM5 | 18.84 | 19.24 | 20.34 | 20.51 |
| CMCC-CMS | 19.47 | 20.26 | 21.14 | 21.10 |
| NorESM1-ME | 19.47 | 19.12 | 20.69 | 20.61 |
| MPI-ESM-MR | 18.51 | 19.10 | 20.19 | 20.06 |
| Average | 18.81 | 19.13 | 20.27 | 20.32 |
| **SST Changes** | **2010** | **2030–2010** | **2060–2010** | **2090–2010** |
| | | **(2030–2010)/Years** | **(2060–2030)/Years** | **(2090–2060)/Years** |
| ACCESS1.3 | 17.83 | 0.70 | 2.95 | 3.16 |
| | | 0.034 | 0.073 | 0.007 |
| CCSM4 | 18.07 | 0.13 | 1.50 | 1.34 |
| | | 0.006 | 0.044 | −0.005 |
| FIO-ESM | 18.40 | −0.04 | −0.40 | 0.02 |
| | | −0.002 | −0.011 | 0.013 |
| CESM1-CAM5 | 18.11 | 0.40 | 1.50 | 1.67 |
| | | 0.019 | 0.036 | 0.005 |
| CESM1-CMS | 18.42 | 0.79 | 1.67 | 1.62 |
| | | 0.038 | 0.028 | −0.002 |
| NorESM1-ME | 18.10 | −0.35 | 1.22 | 1.14 |
| | | −0.017 | 0.051 | −0.003 |
| MPI-ESM-MR | 18.10 | 0.59 | 1.68 | 1.55 |
| | | 0.028 | 0.035 | −0.004 |
| Average | 18.11 | 0.32 | 1.46 | 1.51 |
| | | 0.015 | 0.037 | 0.002 |

The model validation and modification of this paper is relative simple. We use the difference between the CMIP5 model and the WOA data in 2010 as the model error and substrate these errors from CMIP5 model data to modify their projection results. We should use more sophisticated methods with all the model assumptions and conditions to perform model modification. However, it is not an easy task especially for the future projection with many unknown conditions. We donot know whether these errors will remain constant for the 2010–2100 period. Thus. we include the experimental results without projection modification for comparison in this part. The decadal and seasonal SST changes without modification of model projection results are shown in Table A1 and Table A2, respectively. Comparing Table A1 to Table 3, and Table A2 to Table 4, we find that the bottom part of these tables are the same with each other. That is, the investigation of 35 the SST changes in the future using the differences of 2030, 2060, and 2090 relative to that in 2010 is not influenced by the 36 model validation and modification method in this paper.

**Table A2.** The projection results of seasonal SST variations in the next 100 years using seven CMIP5 models and their average result using the raw CMIP5 model data without projection result modification.

| SST | 2010 | | | | 2030 | | | | 2060 | | | | 2090 | | | |
|---|---|---|---|---|---|---|---|---|---|---|---|---|---|---|---|---|
| | Spr. | Sum. | Aut. | Win. | Spr. | Sum. | Aut. | Win. | Spr. | Sum. | Aut. | Win. | Spr. | Sum. | Aut. | Win. |
| ACCESS1.3 | 12.55 | 24.86 | 20.28 | 13.00 | 12.89 | 19.43 | 21.14 | 14.09 | 15.81 | 22.33 | 23.25 | 15.16 | 15.13 | 22.54 | 24.15 | 15.57 |
| CCSM4 | 14.41 | 25.34 | 22.79 | 13.16 | 14.43 | 24.98 | 23.51 | 13.31 | 16.02 | 26.24 | 24.47 | 14.95 | 15.81 | 26.27 | 24.62 | 14.36 |
| FIO-ESM | 17.40 | 25.51 | 23.48 | 15.54 | 16.82 | 25.87 | 23.67 | 15.39 | 15.98 | 25.62 | 24.20 | 14.56 | 16.91 | 26.35 | 23.47 | 15.26 |
| CESM1-CAM5 | 14.27 | 23.98 | 22.92 | 14.17 | 15.12 | 24.55 | 23.07 | 14.20 | 16.43 | 25.70 | 24.03 | 15.21 | 15.72 | 25.88 | 25.07 | 15.36 |
| CMCC-CMS | 15.46 | 23.88 | 23.52 | 15.03 | 16.78 | 24.69 | 23.78 | 15.78 | 17.52 | 25.92 | 24.66 | 16.47 | 17.52 | 25.67 | 24.30 | 16.89 |
| NorESM1-ME | 15.49 | 25.63 | 23.55 | 13.20 | 15.76 | 25.59 | 22.87 | 12.27 | 16.89 | 27.54 | 24.41 | 13.93 | 16.91 | 27.36 | 19.52 | 12.13 |
| MPI-ESM-MR | 14.06 | 23.53 | 23.25 | 13.20 | 14.76 | 24.70 | 22.92 | 14.00 | 16.57 | 24.94 | 24.23 | 15.04 | 15.67 | 25.48 | 24.27 | 14.82 |
| Average | 14.76 | 23.81 | 22.83 | 13.85 | 15.17 | 24.25 | 22.99 | 14.11 | 16.43 | 25.45 | 24.18 | 15.01 | 16.20 | 25.63 | 24.30 | 15.14 |
| **SST Changes** | **2010** | | | | **2030–2010** | | | | **2060–2010** | | | | **2090–2010** | | | |
| | Spr. | Sum. | Aut. | Win. | Spr. | Sum. | Aut. | Win. | Spr. | Sum. | Aut. | Win. | Spr. | Sum. | Aut. | Win. |
| ACCESS1.3 | 12.55 | 24.86 | 20.28 | 13.00 | 0.35 | 0.52 | 0.86 | 1.09 | 3.27 | 3.41 | 2.97 | 2.16 | 2.58 | 3.63 | 3.87 | 2.57 |
| CCSM4 | 14.41 | 25.34 | 22.79 | 13.16 | 0.02 | −0.36 | 0.72 | 0.15 | 1.61 | 0.90 | 1.68 | 1.79 | 1.40 | 0.93 | 1.82 | 1.20 |
| FIO-ESM | 17.40 | 25.51 | 23.48 | 15.54 | −0.57 | 0.36 | 0.20 | −0.15 | −1.42 | 0.11 | 0.72 | −0.99 | −0.49 | 0.84 | 0.00 | −0.28 |
| CESM1-CAM5 | 14.27 | 23.98 | 22.92 | 14.17 | 0.85 | 0.57 | 0.15 | 0.03 | 2.16 | 1.72 | 1.10 | 1.04 | 1.45 | 1.90 | 2.14 | 1.19 |
| CMCC-CMS | 15.46 | 23.88 | 23.52 | 15.03 | 1.33 | 0.81 | 0.26 | 0.75 | 2.07 | 2.04 | 1.15 | 1.45 | 2.06 | 1.79 | 0.79 | 1.86 |
| NorESM1-ME | 15.49 | 25.63 | 23.55 | 13.20 | 0.27 | −0.05 | −0.68 | −0.93 | 1.39 | 1.91 | 0.87 | 0.73 | 1.42 | 1.59 | 0.67 | 0.89 |
| MPI-ESM-MR | 14.06 | 23.53 | 23.25 | 13.20 | 0.70 | 1.17 | −0.33 | 0.80 | 2.51 | 1.41 | 0.98 | 1.84 | 1.61 | 1.95 | 1.03 | 1.62 |
| Average | 14.76 | 23.81 | 22.83 | 13.85 | 0.41 | 0.44 | 0.16 | 0.26 | 1.67 | 1.64 | 1.35 | 1.16 | 1.45 | 1.82 | 1.47 | 1.29 |
| **SST Changes ratios** | **2010** | | | | **2030–2010** | | | | **2060–2030** | | | | **2090–2060** | | | |
| | Spr. | Sum. | Aut. | Win. | Spr. | Sum. | Aut. | Win. | Spr. | Sum. | Aut. | Win. | Spr. | Sum. | Aut. | Win. |
| ACCESS1.3 | 12.55 | 24.86 | 20.28 | 13.00 | 0.017 | 0.025 | 0.041 | 0.052 | 0.094 | 0.093 | 0.068 | 0.035 | −0.022 | 0.007 | 0.029 | 0.013 |
| CCSM4 | 14.41 | 25.34 | 22.79 | 13.16 | 0.001 | −0.017 | 0.034 | 0.007 | 0.051 | 0.041 | 0.031 | 0.053 | −0.007 | 0.001 | 0.005 | −0.019 |
| FIO-ESM | 17.40 | 25.51 | 23.48 | 15.54 | −0.027 | 0.017 | 0.009 | −0.007 | −0.027 | −0.008 | 0.017 | −0.027 | −0.030 | 0.024 | −0.023 | 0.023 |
| CESM1-CAM5 | 14.27 | 23.98 | 22.92 | 14.17 | 0.040 | 0.027 | 0.007 | 0.001 | 0.042 | 0.037 | 0.031 | 0.033 | −0.023 | 0.006 | 0.034 | 0.005 |
| CMCC-CMS | 15.46 | 23.88 | 23.52 | 15.03 | 0.063 | 0.039 | 0.013 | 0.036 | 0.024 | 0.040 | 0.029 | 0.022 | −0.001 | −0.008 | −0.012 | 0.013 |
| NorESM1-ME | 15.49 | 25.63 | 23.55 | 13.20 | 0.013 | −0.002 | 0.032 | 0.045 | 0.036 | 0.063 | 0.050 | 0.054 | 0.001 | −0.010 | −0.007 | 0.005 |
| MPI-ESM-MR | 14.06 | 23.53 | 23.25 | 13.20 | 0.034 | 0.056 | −0.016 | 0.038 | 0.058 | 0.008 | 0.042 | 0.034 | −0.029 | 0.017 | 0.002 | −0.007 |
| Average | 14.76 | 23.81 | 22.83 | 13.85 | 0.020 | 0.021 | 0.007 | 0.013 | 0.041 | 0.039 | 0.038 | 0.029 | −0.007 | 0.006 | 0.004 | 0.004 |

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
