# Peer review of "CMIP5-Based Projection of Decadal and Seasonal Sea Surface Temperature Variations in East China Shelf Seas"

_jmse, doi:10.3390/jmse9040367_

Round 1

Reviewer 1 Report

The paper is well written and presents a very interesting topic. I have only some minor comments:

1. Why have the authors used WOA 2013 and not 2018 ? It is always advisable to use the latest version of World Ocean Atlas.

2. it would be nice to include a map of the area with all the place names mentioned in the paper.

3. Fig.2, use the same limits in the colorbar per season for ALL models.

4. Fig.3 same limits in the colorbar per season. Also, the use of Redblue colormap (or any other colormap used for presenting differences) should have been more illustrative.

5. Table 3. "The dacadal " should be "The decadal ".

6. Fig.5, right column, the use of Redblue colormap (or any other colormap used for presenting differences) should have been more illustrative.

Author Response

Dear Editor and Reviewers,

Thank you very much for your comments, suggestions, and consideration of our paper entitled “CMIP5 based Projection of Decadal and Seasonal Sea Surface Temperature Variations in East China Shelf Seas” (Manuscript Number: jmse-09-00135). We have revised the manuscript according to the reviewers’ comments. Below please find our itemized responses. We also highlight the corresponding changes in the revised manuscript. We hope that this revised manuscript is now acceptable for publication in Journal of Marine Science and Engineering.

Best regards,

Cuicui Zhang

Comments and Suggestions of Reviewer #1:

The paper is well written and presents a very interesting topic. I have only some minor comments:

Comment 1-1: Why have the authors used WOA 2013 and not 2018 ? It is always advisable to use the latest version of World Ocean Atlas.

Response: Thank you very much for this good suggestion. We have changed the validation data from WOA 2O13 to 2018 and modified the manuscript as follows: (1) for all the mentions and descriptions of WOA data, we have changed it from ‘WOA13 V2’to ‘WOA18’; (2) we performed model validation and all the experiments again and modified all the related tables and figures, including Tables 2-4, Figures 2-6; (3) all the analysis related to the results in these tables have been updated; Through comparison to the experimental results before (using WOA13 as validation data), very slight changes (about 0.01) have been founded on Tables 2-4 and there are almost no changes on the bottom part of Table 3-4. Thus, these modifications have no influences on the findings and conclusion of this paper. These modifications have been highlighted in the manuscript.

Comment 1-2: it would be nice to include a map of the area with all the place names mentioned in the paper.

Response: Thank you very much for this comment. We have modified Figure 1 (on Page 6) according to this comment.

Comment 1-3: Fig.2, use the same limits in the colorbar per season for ALL models.

Response: Thank you very much for this comment. We have modified Fig.2 (on Page 8) according to this comment.

Comment 1-4: Fig.3 same limits in the colorbar per season. Also, the use of Redblue colormap (or any other colormap used for presenting differences) should have been more illustrative.

Response: Thank you very much for this comment. We have modified Fig.3 (on Page 9) according to this good suggestion.

Comment 1-5: Table 3. "The dacadal " should be "The decadal ".

Response: Thank you very much for this comment. We have modified this typo on Page 10.

Comment 1-6: Fig.5, right column, the use of Redblue colormap (or any other colormap used for presenting differences) should have been more illustrative.

Response: Thank you very much for this comment. We have modified Fig.5 (on Page 11) and Fig.6 (on Page 14) according to this comment.

Reviewer 2 Report

The authors of the manuscript address an interesting and novel topic, the projection of SST changes in climate scenarios in the long term. Experiments are well designed and results are consistent. I just have one major concern that needs to be addressed in the authors' response. In lines 197 to 200, the authors explain that they calculate "model error" by comparing model results and WOA for 2010. I would call model validation rather than model error. Then, they use this "model error" to modify model projection results before analyzing SST trends and changes. Is this a correct methodology? Is it supported by other authors in other research papers? Do the authors assume the "model error" remains constant for the 2010-2100 period? Is it realistic? A comparison between modified vs not modified model SST changes would be worth publishing.

If these questions are properly answered I would recommend the manuscript for publication.

Minor comments:

  • Please, add some names and country borders in figure 1 to better locate the study area for non Chinese readers
  • Figure 2: Do not repeat non informative titles in each map
  • WOA average period is 2005-2012, compared to 2006-2015. Why not the same period?

Author Response

Dear Editor and Reviewers,

Thank you very much for your comments, suggestions, and consideration of our paper entitled “CMIP5 based Projection of Decadal and Seasonal Sea Surface Temperature Variations in East China Shelf Seas” (Manuscript Number: jmse-09-00135). We have revised the manuscript according to the reviewers’ comments. Below please find our itemized responses. We also highlight the corresponding changes in the revised manuscript. We hope that this revised manuscript is now acceptable for publication in Journal of Marine Science and Engineering.

Best regards,

Cuicui Zhang

Comments and Suggestions of Reviewer #2

Comment 2-1:The authors of the manuscript address an interesting and novel topic, the projection of SST changes in climate scenarios in the long term. Experiments are well designed and results are consistent. I just have one major concern that needs to be addressed in the authors’ response. In lines 197 to 200, the authors explain that they calculate “model error” by comparing model results and WOA for 2010. I would call model validation rather than model error. Then, they use this “model error” to modify model projection results before analyzing SST trends and changes. Is this a correct methodology? Is it supported by other authors in other research papers? Do the authors assume the “model error” remains constant for the 2010-2100 period? Is it realistic? A comparison between modified vs not modified model SST changes would be worth publishing.

Response: Thank you very much for this comment and good suggestion. The model validation and modification of this paper is really simple. We just use the difference between the CMIP5 model and the WOA data in 2010 as the model error and substrate these ‘errors’ from CMIP5 model data to modify their projection results. Actually, we should say this as ‘modification of projection results of CMIP5 model’ other than ‘model modification’. We should use more sophisticated methods with all the model assumptions and conditions to perform model modification. However, it is not an easy task especially for the future projection with many unknown conditions. We also donot know whether these ‘errors’ will remain constant for the 2010-2100 period. What we can do is to use the historical data to consume the future condition based on what we have. You also give us a very good idea that we should include the results without projection result modification for comparison. As the model validation and modification is not the very key issue we want to investigate in this paper, we include this part in the Appendix section on Pages 19-20 as supplementary material. We show the decadal and seasonal SST changes in Table 5 and Table 6, respectively. Comparing Table 5 to Table 3, and Table 6 to Table 4, we find that the bottom part of these tables are the same with each other. That is, the investigation of the SST changes in the future using the differences of 2030, 2060, and 2090 relative to that in 2010 is not influenced by the model validation and modification part. Since the climate change is really what we want to analyze and investigate in this paper, the findings and conclusion of this paper is still convincible.

Comment 2-2: Please, add some names and country borders in figure 1 to better locate the study area for non Chinese readers

Response: Thank you very much for this comment. We have modified Figure 1 (on Page 6) by adding all the place names mentioned in this paper.

Comment 2-3: Figure 2: Do not repeat non informative titles in each map

Response: Thank you very much for this comment. We have modified Fig.2 (on Page 8) according to this comment.

Comment 2-4: WOA average period is 2005-2012, compared to 2006-2015. Why not the same period?

Response: Thank you very much for this comment. First, we have changed the WOA 13 data (averaged over 2005-2012) to the latest version WOA 18 (averaged over 2005-2017), which covers the period of 2006-2015. Second, since the WOA18 data access portal (https://www.ncei.noaa.gov/access/world-ocean-atlas-2018/bin/woa18.pl) only supplies the statistical mean value over 2005-2017 other than the data per year, we can only download and use this prepared data without more calculation of average over 2006-2015.   
